

# Combined effects of dietary faba bean water extract and vitamin K3 on growth performance, textural quality, intestinal characteristics, oxidative and immune responses in grass carp

Yichao Li[1,*], Bin Chen[1,*], Junming Zhang[1], Guangjun Wang[1], Wangbao Gong[1], Jingjing Tian[1], Hongyan Li[1], Kai Zhang[1], Yun Xia[1], Zhifei Li[1], Jun Xie[1] and Ermeng Yu[1,2]

[1] Pearl River Fisheries Research Institute of CAFS, Guangzhou, Guangdong, China
[2] Guangxi Academy of Fishery Sciences, Nanning, Guangxi, China
* These authors contributed equally to this work.

## ABSTRACT

Faba bean water extract (FBW) and vitamin K3 (VK3) have been demonstrated to improve the muscle textural quality of fish. To better apply these two feed additives in commercial aquaculture setting, four experimental diets (control, commercial feed group; 15% FBW, 15% faba bean water extract group; 2.5% VK3, 2.5% vitamin K3 group; combined group, 15% faba bean water extract + 2.5% vitamin K3 group) were formulated to explore their combined effects of FBW and VK3 on the growth, health status, and muscle textural quality of grass carp. The growth performance, textural quality, intestinal characteristics, and oxidative and immune responses were analyzed on days 40, 80 and 120. The results showed that supplementation with higher doses of FBW and VK3 have no influence on growth-related parameters and immune parameters of grass carp. Notably, compared with the control, fish in the combined group had the highest textural qualities (hardness, chewiness and adhesiveness), followed by those in 15% FBW and 2.5% VK3 groups ($P < 0.05$). Also, FBW and VK3, to some extent, may lower antioxidative ability of grass carp, as illustrated by lower levels of GSH and CAT in 15% FBW, 2.5% VK3, and combined groups on day 120 ($P < 0.05$). In addition, enhanced lipase activity was observed in the 15% FBW group. Taken together, the combined supplementation of FBW and VK3 was demonstrated to be a more advanced option than their individual supplementation in a commercial setting owing to the resulting combined effects on both the textural quality and health status of grass carp.

# INTRODUCTION

Aquaculture, the fastest-growing cultured food sector, has become a vital contributor to high-quality animal protein for humans over the past 40 years (*FAO, 2022*). To meet the

Corresponding authors
Jun Xie, xiejunhy01@126.com
Ermeng Yu, boyem34@hotmail.com

growing demand for fish, intensive aquaculture at high stocking density has been adopted as the most common approach for optimizing fish production in aquaculture ecosystems. However, reductions in texture and flesh quality have become one of the most important issues facing aquaculture owing to environmental factors (*e.g.*, poor water quality and overcrowding stress) and feeding history (*e.g.*, excessive dietary carbohydrates) in high-density cultivation modes, ultimately leading to poor consumer acceptance and considerable economic losses for producers (*Wu et al., 2018*; *Roth, Slinde & Arildsen, 2006*; *Wu et al., 2021*). Thus, improving fish muscle quality for enhancing aquaculture efficiency and profitability is required.

Texture is one of the most important quality characteristics of fish products in terms of consumer acceptance (*Chen et al., 2021*). According to some studies, supplementation with functional additives is a feasible approach to improve textural quality of fish (*Wu et al., 2021*; *Ma et al., 2020*). Among these, dietary faba bean water extract (FBW) at 12.69 g kg$^{-1}$ effectively improved the textural quality and growth performance of ordinary grass carp (*Ma et al., 2020*). A similar textural improvement was found in Nile tilapia fed FBW supplemented in the expanded pellet diet and pellet diet (*Li et al., 2022*). Notably, vicine (a pro-oxidant), the active substances in FBW, may be responsible for the improvement in the textural quality of grass carp and tilapia (*Ma et al., 2020*). In our previous study, dietary vitamin K3 (VK3, a dietary oxidant) at 20 g kg$^{-1}$ was demonstrated to be more effective for muscle texture improvement in grass carp than faba bean (*Chen et al., 2021*). However, at least 3 months is required for FBW and VK3 to achieve the expected effect on textural quality improvement, which increases the expenditures and farming risk of fish farmers in commercial settings. The beneficial effects of most feed additives are dose-dependent, and if optimal doses exceed these benefits, they might have side effects on the growth and health of fish (*Doan et al., 2020*; *Jiang et al., 2016*; *Yousefi et al., 2021*). Thus, the doses of FBW and VK3 must be increased to shorten the farming period and explore whether they are associated with side effects that impact the growth and health of fish. Such findings would contribute to more effective application of these feeding protocols in aquaculture by improving profitability, and enabling higher efficiency and lower farming risk.

According to some studies, the combination of two feed additives with similar biological functions has better functional effects than single additive, owing to combined effects (*Khan et al., 2017*; *Al-Deriny et al., 2020*; *Shalata et al., 2021*). Thus, the combined effects of the two feed additives should be explored owing to their ability to interaction, especially for fish in aquatic environments. Therefore, we hypothesised that, as both feed additives aid in textural quality improvement, their combined supplementation might have better combined effects on the textural improvement of fish than their individual supplementation.

Grass carp (*Ctenopharyngodon idellus*), one of the most important freshwater aquaculture species, is ranked first in the world with a global production of approximately six million tons in 2020 (*FAO, 2022*). Thus, the present study aimed to: (1) evaluate the effects of higher doses of FBW and VK3 (15% FBW and 2.5% VK3) on textural quality and muscle structure of grass carp; and (2) determine the combined effects (15% FBW + 2.5% VK3) of FBW and VK3 on the growth, health status, and muscle textural quality of grass

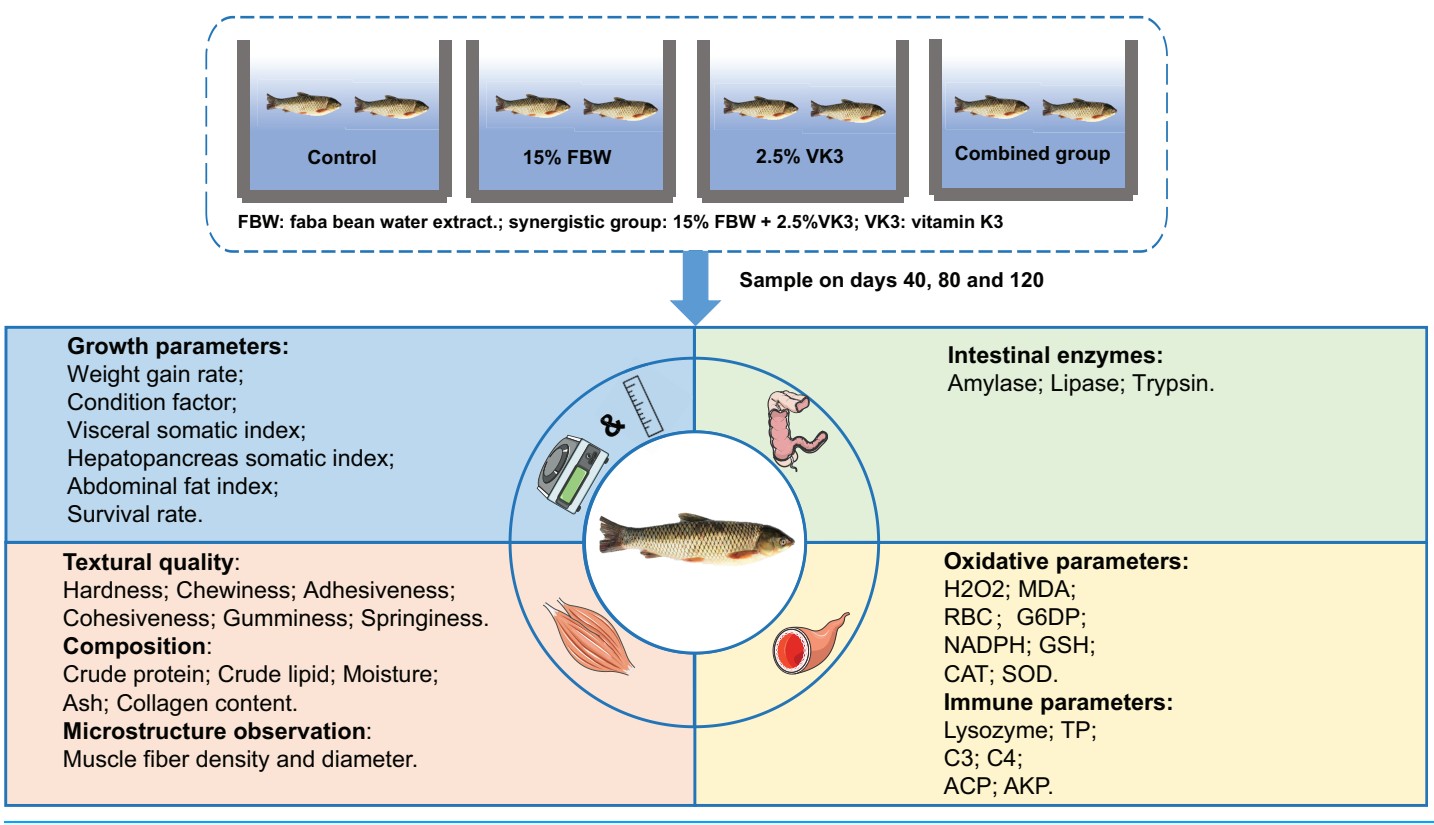

**Figure 1** **The experimental process.**

carp. The growth performance, intestinal characteristics, and oxidative and immune responses of the fish were also examined.

## MATERIALS AND METHODS

### Experimental diets

The experimental process is illustrated in Fig. 1. The formulations and chemical compositions of the experimental diets are listed in Table 1. FBW (crude protein, 37.4%; crude fat, 0.5%; vicine, 2.5%) was obtained according to the methods described in a previous study (*Ma et al., 2020*). Fish in the control group were fed a commercial feed containing the ingredients listed in Table 1. The 15% FBW group were fed feed containing 15% FBW according to a previous study (*Ma et al., 2020*). The combined group was fed feeds containing 15% FBW and 2.5% VK3. The 2.5% VK3 group was fed commercial feed containing VK3 (25 g/kg) (2-Methyl-1,4-naphthoquinone; Macklin, 98%), which was determined based on the content of vicine (2.5%) in the 15% FBW group (*Ma et al., 2020*; *Chen et al., 2021*).

### Fish culture

Three hundred grass carp were obtained from a fish farm in Guangzhou, China. For acclimation, the grass carp were stocked in a concrete tank (3 m × 3 m × 2 m) and fed commercial feed for 15 d. Then, a total of 180 grass carp were allocated to 18 cement pools

**Table 1 The ingredients and proximal composition of experimental diets.**

|  | Control | 15% FBW | 2.5% VK3 | Combined group |
|---|---|---|---|---|
| FBW (g/kg) | 0 | 150 | 0 | 150 |
| Vitamin K3 (g/kg) | 0 | 0 | 25 | 25 |
| Fish meal (g/kg) | 30 | 30 | 30 | 30 |
| Chicken bone meal (g/kg) | 30 | 30 | 30 | 30 |
| Soybean meal (g/kg) | 250 | 141.7 | 258.3 | 154.2 |
| Rapeseed meal (g/kg) | 220 | 220 | 220 | 220 |
| Fine rice bran (g/kg) | 30 | 30 | 30 | 30 |
| Wheat meal (g/kg) | 360 | 318.3 | 326.7 | 280.8 |
| Soybean oil (g/kg) | 30 | 30 | 30 | 30 |
| Monocalcium phosphate (g/kg) | 20 | 20 | 20 | 20 |
| Bentonite (g/kg) | 20 | 20 | 20 | 20 |
| Vitamin and Mineral (g/kg)[a] | 10 | 10 | 10 | 10 |
| Total | 1,000 | 1,000 | 1,000 | 1,000 |
| Feeding amounts | 100% | 100% | 100% | 100% |
| Chemical composition |  |  |  |  |
| Crude protein (%) | 28.86 | 28.83 | 28.79 | 28.89 |
| Crude fat (%) | 5.08 | 4.98 | 5.04 | 4.94 |

Notes:
[a] Contained 1% vitamin, 1% mineral, and 1% limestone carrier; ingredients including (kg$^{-1}$): VA 4000 IU, VD3 800 IU, VE 50 IU, VB1 2.5 mg, VB2 9 mg, VB6 10 mg, VC 250 mg, nicotinic acid 40 mg, pantothenic acid calcium 30 mg, biotin 100 μg, betaine 1,000 mg, Fe 140 mg, Cu 2.5 mg, Zn 65 mg, Mn 19 mg, Mg 230 mg, Co 0.1 mg, I 0.25 mg, Se 0.2 mg.
Control, commercial feed group; 15% FBW, 15% faba bean water extract group; 2.5% VK3, 2.5% vitamin K3 group; Combined group, 15% faba bean water extract + 2.5% vitamin K3 group.

(2.5 m × 2.5 m × 1.2 m) (three replicates per group and 10 fish per pool). The feeding trial was conducted for 120 d at the Pearl River Fisheries Research Institute (Guangdong, China). There was no difference in the initial weight (251 ± 6 g) among the groups ($P > 0.05$). Fish were fed at 08:00 and 16:00 every day, with a daily feeding amount of 2–4% of grass carp body weight. The experimental conditions remained the same in all pools: pH 6.6–7.6, dissolved oxygen >7.0 mg/L, and water temperature 23–28 °C.

## Sample collection

After fasting for 24 h, six fish from each group were randomly captured on days 40, 80, and 120 and were euthanized with pH-buffered tricaine methanesulfonate (250 mg/L) in large plastic containers (1 m × 1 m × 0.5 m). After fin and operculum movement ceased, the body weight and body length of the fish were measured to determine growth performance. The final weights of grass carp are shown in File S1. Fish were sampled from each group. Blood samples (10 mL) were collected *via* tail vein of each fish. Three milliliters of blood was collected in an anticoagulant tube containing EDTAK2 for hematological analyses, and the remaining 7 mL was left standing for 5 h at 4 °C. Subsequently, the blood samples were centrifuged at 3,500 rpm for 10 min at 4 °C to obtain serum samples, which were then stored at −80 °C until analysis of the biochemical indexes (oxidative, antioxidative and immune parameters). The viscera, hepatopancreas, and abdominal fat were weighted to calculate growth-related parameters (visceral somatic index, hepatopancreas somatic

index, and abdominal fat index). Some muscles (3 mm$^3$ × 3 mm$^3$ × 3 mm$^3$) of the fifth dorsal fin and lateral line were sampled for collagen content measurement and and haematoxylin and eosin staining. Another part of the back muscles (1 cm × 1 cm × 0.5 cm) was collected for textural parameters. The intestinal samples were collected for analysis of the intestinal enzymes.

Growth-related parameters were calculated as follows.

Weight gain rate (WGR, %) = (final weight−initial weight)/initial weight × 100

Condition factor (CF, %) = body weight/length3 × 100

Visceral somatic index (VSI, %) = visceral weight/body weight × 100

Hepatopancreas somatic index (HSI, %) = hepatopancreas weight/body weight × 100

Feed conversion rate (FCR) = total food intake/(final weight − initial weight)

Abdominal fat index (AFI, %) = abdominal fat weight/body weight × 100

Survival rate (SR, %) = the number of surviving fish/total initial number of fish × 100

The experimental protocols used in this study were approved by the Laboratory Animal Ethics Committee of Pearl River Fisheries Research Institute, CAFS, China, under permit number LAEC-PRFRI-2021-03-03.

## Muscle textural analysis

A Universal TA Texture Analyzer was used to measure muscle textural parameters (including hardness, chewiness, gumminess, adhesiveness, cohesiveness and springiness) (Tengba, China) (*Chen et al., 2021*). The collagen content of he muscle was determined using an Ultra Sensitive Fish ELISA Kit (Sino, Beijing, China) (Kit No. YX-E21992F).

## Measurement of the biochemical indexes in serum

Red blood cell (RBC) count was measured as previously described (*Natt & Herrick, 1952*), and counting was conducted using a light microscope (Olympus, Tokyo, Japan). Serum oxidative and antioxidative parameters were measured using Ultra Sensitive Fish ELISA Kits (Sino, Beijing, China), including glucose-6-phosphate dehydrogenase (G6DP) (Kit No. YX-E21988F), photohydrogen peroxide ($H_2O_2$) (Kit No. YX-E21807F), malondialdehyde (MDA) (Kit No. YX-E21969F), superoxide dismutase (SOD) (Kit No. YX-E22111F), catalase (CAT) (Kit No. YX-E22107F), glutathione (GSH) (Kit No. YX-E21817F), and nicotinamide adenine dinucleotide (NADPH) (Kit No. YX-E22078F). Serum immune parameters were also measured using Ultra Sensitive Fish ELISA Kits (Sino, Beijing, China), including lysozyme (Kit No. YX-E21980F), total protein (TP) (Kit No. YX-E21984F), complement C3 and C4 (Kit No. YX-E21981F and Kit No. YX-E21983F), alkaline phosphatase (AKP) (Kit No. YX-E21952F), and acid phosphatase (ACP) (Kit No. YX-E22084F).

**Table 2 Growth parameters.**

|          | Control         | 15% FBW         | 2.5% VK3        | Combined group   |
|----------|-----------------|-----------------|-----------------|------------------|
| WGR (%)  | 217.26 ± 1.94   | 208.92 ± 1.97   | 207.74 ± 1.44   | 213.33 ± 2.17    |
| CF (%)   | 17.27 ± 0.51    | 17.87 ± 0.64    | 18.12 ± 0.62    | 17.65 ± 0.68     |
| VSI (%)  | 14.31 ± 0.41    | 13.22 ± 0.34    | 13.88 ± 0.38    | 13.69 ± 0.65     |
| HSI (%)  | 2.35 ± 0.29     | 2.22 ± 0.22     | 2.16 ± 0.18     | 2.27 ± 0.3       |
| FCR      | 1.89 ± 0.13     | 1.98 ± 0.15     | 1.97 ± 0.11     | 1.91 ± 0.12      |
| AFI (%)  | 2.79 ± 0.48     | 2.42 ± 0.26     | 2.52 ± 0.32     | 2.41 ± 0.33      |
| SR (%)   | 100             | 100             | 100             | 100              |

**Note:**

Control, commercial feed group; 15% FBW, 15% faba bean water extract group; 2.5% VK3, 2.5% vitamin K3 group; combined group, 15% faba bean water extract + 2.5% vitamin K3 group; WGR, weight gain rate; CF, condition factor; VSI, visceral somatic index; HSI, hepatopancreas somatic index; FCR, feed conversion rate; AFI, abdominal fat index; SR, survival rate; values of the same column with different letters were significantly different ($n = 6$, $P < 0.05$).

## Measurement of enzyme activities in the intestine

The intestine samples were thawed, chopped, and weighed. Thereafter, the sections were mixed with an appropriate amount of phosphate-buffered saline (PBS, pH 7.2–7.4) and homogenized using a fully automatic rapid grinder on ice. After centrifugation at 2,500 rpm for 20 min at 4 °C, the supernatant was collected to measure the oxidative indexes and enzyme activities in the intestine. The intestinal enzyme activities (trypsin, amylase and lipase) were measured using respective Ultra Sensitive Fish ELISA Kits (Kit No. YX-E21965F, Kit No. YX-E21813F and Kit No. YX-E21975F).

## Microstructure observation of the muscle

H&E staining of the muscle tissues was performed as previously reported (*Ma et al., 2020*). The H&E stained slides were employed for morphometric analysis with a light microscopy BX51 (Olympus, Tokyo, Japan). Assuming that the muscle fibers are cylindrical, the muscle diameter was calculated using the following: equation $s = \pi r^2$ (where r and s represent the radius and muscle fiber area, respectively). The morphological differences in the muscle were observed using ImageJ (National Institutes of Health, Bethesda, MD, USA), including muscle fiber diameter (mf), muscle fiber density, and matrix between muscle fibers (mmf). A total of 500 muscle fibers were measured in each sample.

## Data analysis

Statistical analyses were performed using GraphPad Prism 7.0 (GraphPad Software, San Diego, CA, USA), and data were analyzed by one-way analysis of variance (ANOVA) followed by Duncan's test. A *P* value less than 0.05 was considered to indicate significant difference. The results are presented as mean ± SE. Statistical results are fully reported in File S1, including degrees of freedom, the exact F values, effect size and *P*-value.

# RESULTS

## Growth performance

The growth performance of fish in the four groups is shown in Table 2. There was no significant difference among all growth-related parameters, including weight gain rate

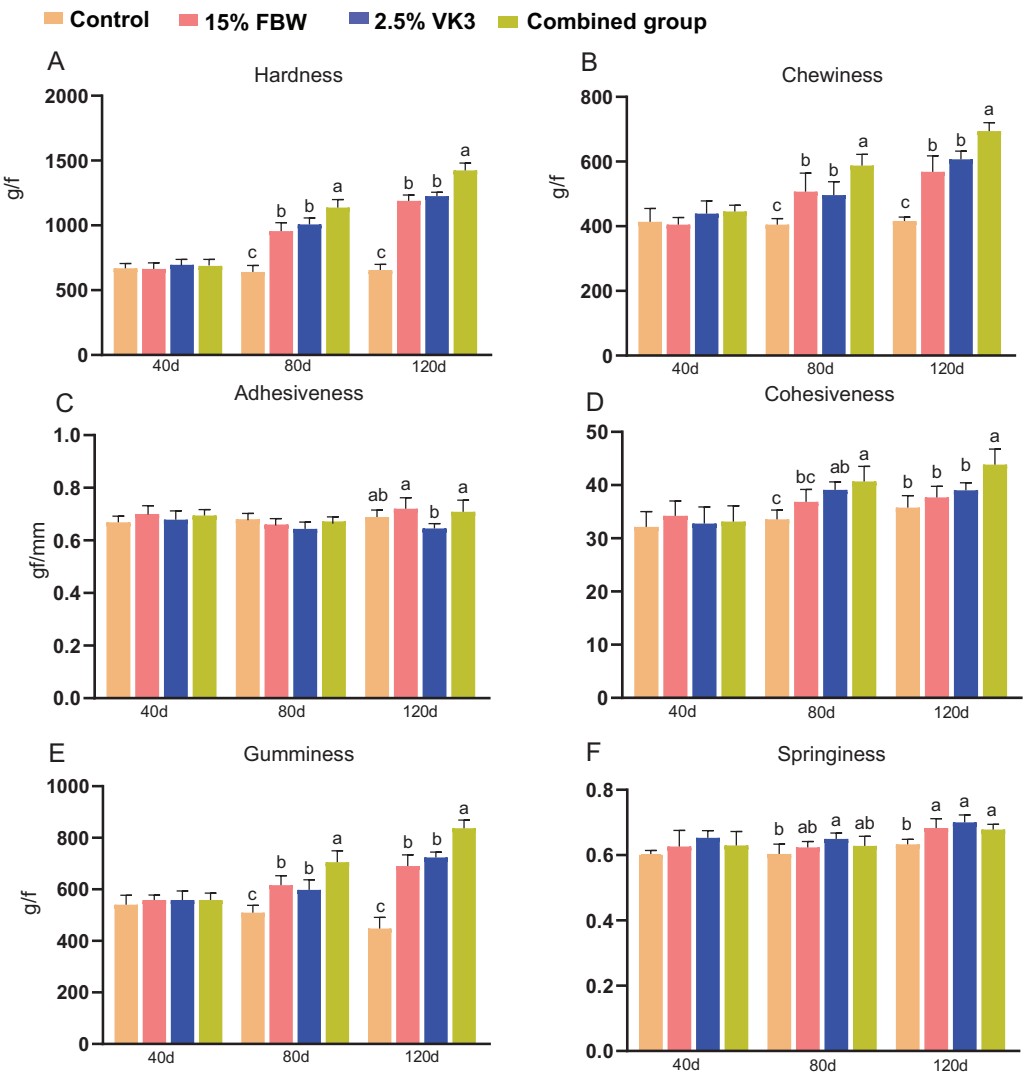

**Figure 2 Muscle textural quality on days 40, 80, and 120.** Control, commercial feed group; 15% FBW, 15% faba bean water extract group; 2.5% VK3, 2.5% vitamin K3 group; combined group, 15% faba bean water extract + 2.5% vitamin K3 group. (A) Hardness; (B) chewiness; (C) adhesiveness; (D) cohesiveness; (E) gumminess; (F) springiness. Different lowercase letters were significantly different ($P < 0.05$).

(WGR), condition factor (CF), visceral somatic index (VSI), hepatopancreas somatic index (HSI), feed conversion rate (FCR), abdominal fat index (AFI), survival rate (SR) ($P > 0.05$).

## Textural parameters and microstructure observation of the muscle

The textural parameters of the muscles (hardness, chewiness, adhesiveness, gumminess, cohesiveness, and springiness) were measured for the four groups (Fig. 2). There was no significant difference in textural parameters among the four groups on day 40 (Fig. 2). The hardness, chewiness, and gumminess of the control group were lower than those of the other three groups on days 80 and 120 ($P < 0.05$) (Figs. 2A, 2B and 2E), and the hardness, chewiness, and gumminess of the combined group were higher than those of the 15% FBW

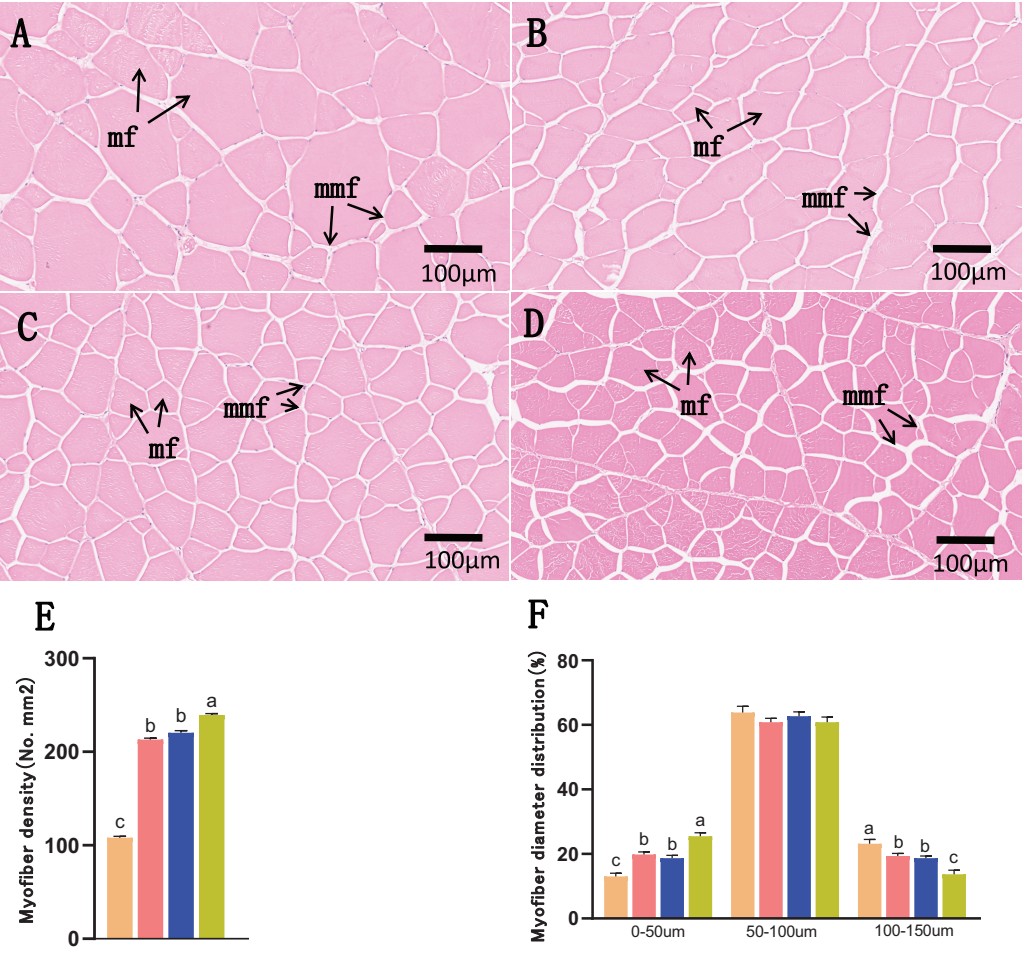

**Figure 3 Muscle transverse section microstructure on day 120.** (A) Commercial feed group; (B) 15% FBW group; (C) 2.5% VK3 group; (D) combined group (15% FBW + 2.5 VK3); (E) myofiber density; (F) myofiber diameter distribution. H&E staining, bar = 100 µm. mf, muscle fiber; mmf, matrix between muscle fibers. Different letters were significantly different ($P < 0.05$).

and 2.5% VK3 groups on days 80 and 120 ($P < 0.05$) (Figs. 2A, 2B and 2E). Furthermore, the combined group showed the highest level of cohesiveness on days 80 and 120 ($P < 0.05$) (Fig. 2D). The springiness of the control group was lower than those of the other three groups on day 120 ($P < 0.05$) (Fig. 2F).

Transverse microstructure diagrams of the muscles on day 120 are shown in Fig. 3. Overall, the muscle fiber densities of the control group were lower than those of the other three groups on day 120 ($P < 0.05$). Notably, muscle fiber densities in the combined group were higher than those in the 15% FBW and 2.5% VK3 groups ($P < 0.05$), whereas no apparent difference was found among the 15% FBW and 2.5% VK3 groups ($P > 0.05$) (Fig. 3E). Consistently, the number of small-diameter fibers (0–50 µm) in the control group was less than those of the other three groups on days 120 ($P < 0.05$), and the number of small-diameter fibers (0–50 µm) in the combined group was higher than those in the 15% FBW and 2.5% VK3 groups ($P < 0.05$), but the number of larger-diameter fibers

**Table 3 Muscle nutritional composition and collagen content.**

|  | Crude protein (%) | Crude fat (%) | Moisture (%) | Ash (%) | Collagen content (mg/g) |
|---|---|---|---|---|---|
| Control | 20.02 ± 0.43 | 2.78 ± 0.24 | 76.78 ± 0.40 | 1.17 ± 0.10 | 8.24 ± 0.48[b] |
| 15% FBW | 19.11 ± 0.42 | 3.01± 0.32 | 76.87 ± 0.42 | 1.16 ± 0.11 | 10.47 ± 0.4[a] |
| 2.5% VK3 | 20.07± 0.52 | 3.05± 0.28 | 77.41 ± 0.25 | 1.19 ± 0.10 | 8.98 ± 0.28[b] |
| Combined group | 19.45 ± 0.48 | 3.12 ± 0.32 | 76.57 ± 0.7 | 1.18 ± 0.11 | 10.96 ± 0.3[a] |

Note:
Control, commercial feed group; 15% FBW, 15% faba bean water extract group; 2.5% VK3, 2.5% vitamin K3 group; 5% VK3, 5% vitamin K3 group; combined group, 15% faba bean water extract + 2.5% vitamin K3 group. Values of the same column with different letters were significantly different ($n$ = 6, $P$ < 0.05).

(100–150 μm) showed opposite trend ($P$ < 0.05) (Fig. 3F). There were no significant differences among 15% FBW and 2.5% VK3 groups in myofiber diameter distribution and density ($P$ > 0.05).

### Muscle nutritional composition and collagen content

The nutritional composition and collagen content of the muscle are shown in Table 3. The 15% FBW and combined groups had higher muscle collagen content than the control and 2.5% VK3 groups ($P$ < 0.05). However, this feature did not significantly differ among the control and 2.5% VK3 groups ($P$ > 0.05).

### Haemocytolysis indexes

To explore the effects of FBW and VK3 on oxidative responses in grass carp, haemolysis indexes, including hydrogen peroxide ($H_2O_2$), malondialdehyde (MDA), red blood cell counts (RBC), glucose-6-phosphate dehydrogenase (G6PD) activity, reduced nicotinamide adenine dinucleotide phosphate (NADPH), glutathione (GSH), catalase (CAT), superoxide dismutase (SOD) activity, were measured (Fig. 4). Compared with the control group, the 15% FBW, 2.5% VK3, and combined groups displayed the higher levels of $H_2O_2$ on day 120 ($P$ < 0.05) (Fig. 4A). Compared with the control group, the 2.5% VK3 group displayed the higher levels of MDA on day 80, whereas the combined group showed the higher levels of MDA on day 120 ($P$ < 0.05) (Fig. 4B). There were no significant differences in RBC among the four groups ($P$ > 0.05) (Fig. 4C). Furthermore, lower levels of G6DP and NADPH were observed in the combined group compared with the control group on day 120 ($P$ < 0.05) (Figs. 4D and 4E). The control group shown lower levels of GSH and CAT than those of 15% FBW, 2.5% VK3, and combined groups on day 120 (Figs. 4F and 4G). In addition, compared with the control group, the 15% FBW, 2.5% VK3, and combined groups displayed the higher levels of SOD on days 80 and 120 ($P$ < 0.05) (Fig. 4H). Overall, these results suggest that FBW and VK3, to some extent, may lead to mild oxidative damage to grass carp blood.

### Immune parameters in serum

The effects of different diets on immune parameters, including lysozyme, acid phosphatase (ACP), alkaline phosphatase (AKP), total protein (TP), complement C3, and complement C4 were determined (Fig. 5). Serum lysozyme activity was significantly decreased in the 15% FBW group compared to the control on day 120 ($P$ < 0.05) (Fig. 5A). The 15% FBW

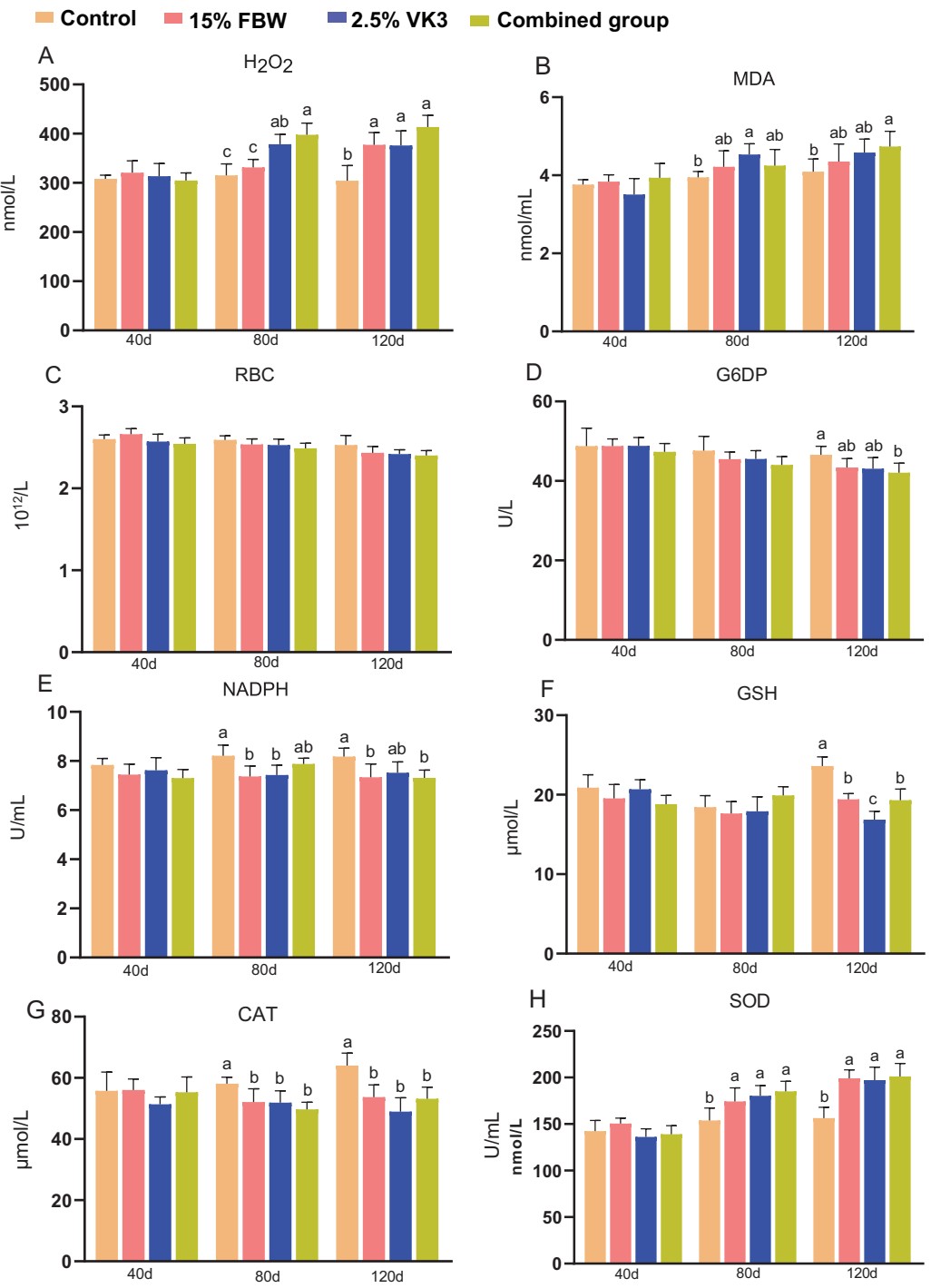

**Figure 4 Hematological indexes, oxidative and antioxidative parameters of serum on days 40, 80 and 120.** Control, commercial feed group; 15% FBW, 15% faba bean water extract group; 2.5% VK3, 2.5% vitamin K3 group; combined group, 15% faba bean water extract + 2.5% vitamin K3 group. (A) Hydrogen peroxide ($H_2O_2$); (B) malondialdehyde (MDA); (C) red blood cell counts (RBC); (D) glucose-6-phosphate dehydrogenase (G6PD) activity; (E) reduced nicotinamide adenine dinucleotide phosphate (NADPH); (F) glutathione (GSH); (G) catalase (CAT); (H) superoxide dismutase (SOD) activity; Different letters were significantly different ($P < 0.05$).

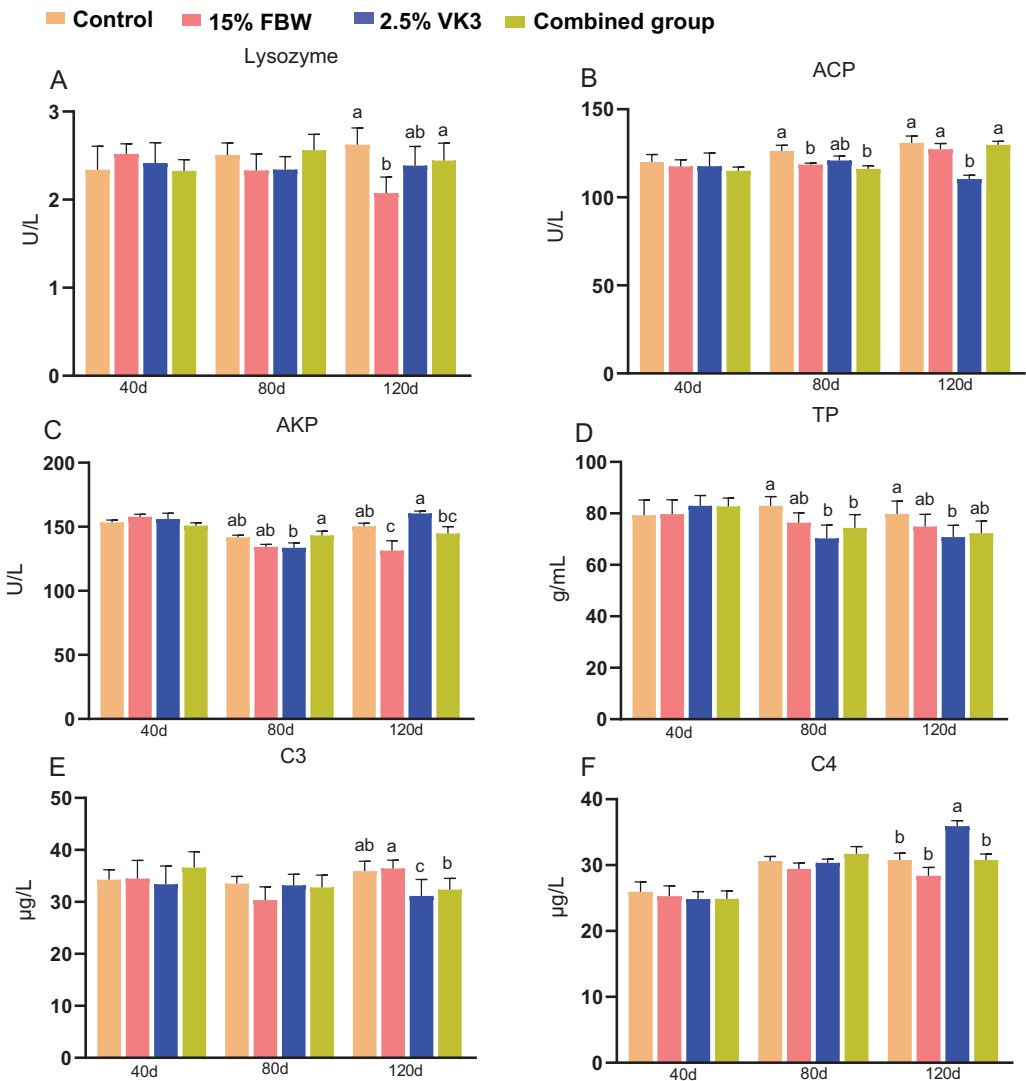

**Figure 5 Immune parameters of serum on days 40, 80, and 120.** Control, commercial feed group; 15% FBW, 15% faba bean water extract group; 2.5% VK3, 2.5% vitamin K3 group; combined group, 15% faba bean water extract + 2.5% vitamin K3 group. (A) Lysozyme; (B) acid phosphatase (ACP); (C) alkaline phosphatase (AKP); (D) total protein (TP); (E) complement C3; (F) complement C4. Different letters were significantly different ($P < 0.05$).

and combined groups had the lower ACP activity than the control on day 80 ($P < 0.05$), while the 2.5%VK3 group exhibited lower ACP activity than the control on day 120 ($P < 0.05$) (Fig. 5B). Notably, the 15% FBW and 2.5% VK3 groups had the lower TP content than the control on day 80 ($P < 0.05$), and 2.5% VK3 group also exhibited lower TP content than the control on day 120 ($P < 0.05$) (Fig. 5D). The 2.5% VK3 group had a lower the C3 contents than the control on day 80, while had a higher the C4 contents than the control on day 80 (Figs. 5E and 5F). Generally, these results indicate that higher doses of FBW and VK3 (5% VK3) exert no negative influence on the immune status of grass carp.

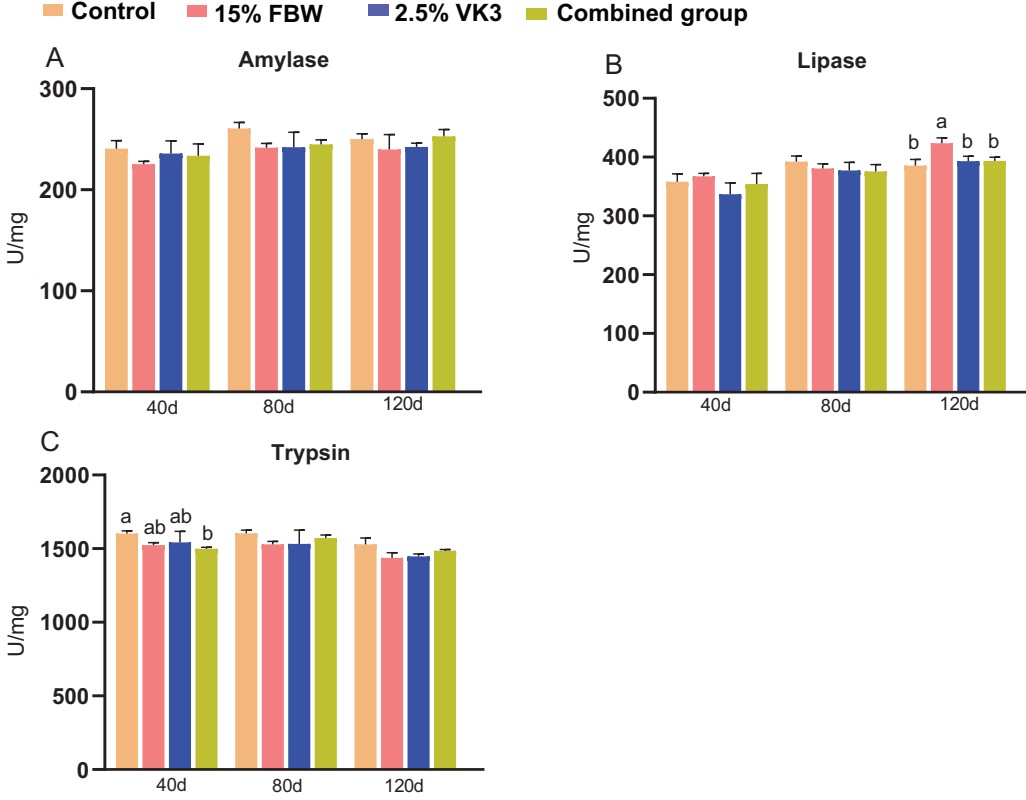

**Figure 6 The intestinal digestive enzymes on days 40, 80 and 120.** Control, commercial feed group; 15% FBW, 15% faba bean water extract group; 2.5% VK3, 2.5% vitamin K3 group; combined group, 15% faba bean water extract + 2.5% vitamin K3 group. (A) Amylase; (B) lipase; (C) trypsin. Different letters were significantly different ($P < 0.05$).

## Intestinal digestive enzyme activities

To explore the potential influence of the five different feed additives on intestinal health status, we measured the activities of intestinal digestive enzymes (amylase, lipase and trypsin) (Fig. 6). The activities of the three intestinal digestive enzymes were generally not affected by the administration of FBW and VK3 compared with the control (Fig. 6). Specifically, the lipase activity in the 15% FBW group were higher than those in the control group on day 120 ($P < 0.05$) (Fig. 6B). The trypsin activity in the combined group were lower than those in the control group on day 40 ($P < 0.05$) (Fig. 6C).

## DISCUSSION

Previous studies from our laboratory revealed that FBW could be used to improve the textural quality of grass carp and tilapia when employed as an aquafeed additive at dietary levels of approximately 100 g kg$^{-1}$ (*Ma et al., 2020*; *Li et al., 2022*). Our initial studies also found that VK3 could contribute to the improvement in muscle texture of grass carp at dietary levels of 20 g kg$^{-1}$ (*Chen et al., 2021*). However, the effects of higher doses of FBW and VK3 on textural quality improvement deserve further exploration. Such findings would help to improve feeding protocols in the commercial settings by shortening the farming period. Furthermore, as both FBW and VK3 aid in textural quality improvement,

no information is available regarding the combined effects of FBW and VK3 on fish muscle textural improvement. Thus, this study sough to determine the effects of higher doses of FBW and VK3 on the growth performance, textural quality, intestinal characteristics, and oxidative and immune responses of grass carp, and further explore their combined effects on these parameters.

Our earlier studies revealed that FBW had no impact on the growth-related parameters of grass carp and tilapia at relatively low dietary levels (12.5% and 8% respectively) (*Ma et al., 2020*; *Li et al., 2022*). In the present study, diets enriched with FBW in 15% FBW and combined groups exerted no effect on the growth and feeding parameters of grass carp, including WGR, CF, VSI, HSI, FCR, AFI, and SR, which is consistent with our previous study (*Ma et al., 2020*; *Li et al., 2022*). In addition, previous study has reported that the vitamin K requirement and recommendations in fish were below 100 mg kg$^{-1}$ (*Krossøy, Waagbo & Ørnsrud, 2011*). However, in present study, diets enriched with VK3 in 2.5% VK3 and combined groups have no influence on the growth and feeding parameters of grass carp, indicating that higher doses of VK3 would not compromise growth and survival of grass carp. Overall, such findings suggest that higher doses of FBW and VK3 are promising protocols for muscle textural improvement in fish. To apply our findings to a commercial setting, the more appropriate doses of FBW and VK3 must be further determined.

The improvement in muscle textural quality (*e.g.*, hardness, chewiness, and gumminess) is the most notable feature of crisp grass carp which is closely associated with increased muscle fiber density and collagen content (*Yang et al., 2015*; *Yu et al., 2019*). In a previous study, FBW was reported to improve muscle textural quality by enhancing muscle fiber density and collagen content in grass carp and tilapia (*Ma et al., 2020*; *Li et al., 2022*). Consistent with these previous findings, muscle textural quality (including hardness, chewiness, and gumminess), muscle fiber density, and collagen content were increased in the grass carp of the 15% FBW group. Our previous study revealed that VK3 could effectively improve the muscle textural quality of grass carp by increasing the accumulation of ROS in the muscle, which promote the muscle fiber hyperplasia or collagen synthesis (*Chen et al., 2021*; *Yu et al., 2020*). In the present study, VK3 exhibited a similar function in improving textural quality in the 2.5% VK3 group, as found in a previous study (*Chen et al., 2021*). Nevertheless, supplementation with VK3 had no impact on muscle collagen content, but increased the muscle fiber density of grass carp in the 2.5% VK3 group. Combined with the microstructure observation results (increased muscle fibre density in the 2.5% VK3 group), VK3, unlike FBW, can be inferred to improve muscle texture mainly by increasing muscle fire density rather than muscle collagen content. Interestingly, the combined group displayed the best enhancing effect on muscle textural improvement, suggesting that FBW and VK3 might have strong combined interactions and positively potentiated muscle textural improvement. To better apply these findings to commercial settings, their combined effects must be further explored in different combinations.

In our previous study, FBW was demonstrated to induce red cell haemolysis due to decreased levels of G6PD, NADPH, and GSH, and increased contents of $H_2O_2$ in grass

carp and tilapia (*Ma et al., 2020*; *Li et al., 2022*). VK3, an oxidant, induces the generation of $H_2O_2$ and causes red cell haemolysis by oxidizing intracellular hemoglobin to methemoglobin in mammals, accompanied by decreased levels of G6PD and GSH (*Kelly & Newman, 2022*). Therefore, we investigated the effects of FBW and VK3 on these parameters. However, no obvious haemolysis was found in the four groups, as demonstrated by the similar RBC count, indicating that high doses of FBW and VK3 would not result in apparent oxidative damage of red blood cells. Furthermore, antioxidant system is pivotal in fish well-being and is responsible for the protection of fish against oxidative stress. SOD, NADPH, CAT, and GSH are key antioxidant components in the antioxidant system (*Diebold & Chandel, 2016*), and decreased levels of NADPH, CAT, and GSH indicate a reduction in the body's antioxidative ability (*Taheri Mirghaed, Hoseini & Ghelichpour, 2018*). The present results revealed the weakened antioxidative ability of grass carp fed diets supplemented with FBW and VK3 in 15% FBW, 2.5% VK3, and combined groups, as illustrated by the increased levels of $H_2O_2$ and MDA and decreased levels of NADPH and GSH. Therefore, higher doses of FBW and VK3 may cause mild decreased antioxidative ability in blood to some extent.

Innate immune parameters, such as lysozyme, lysosomal enzymes (ACP and AKP), immunoglobulins (TP), and complement (C3 and C4), as the key elements involved in fish immune response, are important indicators of fish health status (*Gomez, Sunyer & Salinas, 2013*). Here, 15% FBW supplementation was found to exert little negative influence on the immunity of grass carp compared to the control, demonstrating by lower levels of lysozyme and C3. Anti-nutritional factors in dietary plant protein ingredients have been reported to negatively impact the immune response in aquatic animals (*Bone, 2013*; *Zhang et al., 2020*). Therefore, we infer that the mildly decreased immunity in the 15% FBW group can be attributed to the presence of anti-nutritional factors in the FBW.

Fish growth and health are closely related to intestinal enzymes activity (amylase, trypsin, and lipase) (*Gao et al., 2010*; *De Santis et al., 2016*). FBW has been reported to enhanced the activity of intestinal amylase and lipase in tilapia (*Li et al., 2022*). Consistent with the previous studies (*Li et al., 2022*), enhanced lipase activity was observed in the 15% FBW group. Taken together, these results suggested that combined supplementation of FBW and VK3 could be a reasonable option in a commercial setting owing to their combined effects on both the health and textural quality of grass carp.

This study had one limitation that the use of only one combination of FBW and VK3 to investigate their combined effects on growth, feeding, health, and muscle textural improvement. Therefore, to obtain better combined effects, other combinations of FBW and VK3 should be further explored. As a next step, we will determine the most optimal individual FBW or VK3 and determine the best combined supplementation.

## CONCLUSION

Overall, FBW improved the muscle textural quality (hardness, chewiness, and gumminess) by increasing the muscle fiber density and collagen content of grass carp in the 15% FBW group. VK3 mainly improved muscle textural quality *via* an increase in muscle fiber density. Interestingly, the combined supplementation of FBW and VK3 in the combined

group had the most optimal effects on muscle textural improvement by increasing muscle fiber density and collagen content, and only had little side effects on immune response and intestinal health of grass carp compared to the control. Therefore, these findings indicate that FBW and VK3 had combined effects on the textural quality and growth performance of grass carp. We will further explore the optimal combined supplementation of these two feed additives.

### Funding

This research project was sponsored by the China Agriculture Research System of MOF and MARA (No. CARS-45-21) and Central Public-Interest Scientific Institution Basal Research Fund (CAFS, NO. 2021XT03). The funders had no role in study design, data collection and analysis, decision to publish, or preparation of the manuscript.

### Grant Disclosures

The following grant information was disclosed by the authors:
China Agriculture Research System of MOF and MARA: CARS-45-21.
Central Public-Interest Scientific Institution Basal Research Fund: CAFS, 2021XT03.

### Competing Interests

The authors declare that they have no competing interests.

### Author Contributions

- Yichao Li conceived and designed the experiments, performed the experiments, analyzed the data, prepared figures and/or tables, and approved the final draft.
- Bin Chen performed the experiments, analyzed the data, prepared figures and/or tables, and approved the final draft.
- Junming Zhang performed the experiments, analyzed the data, prepared figures and/or tables, and approved the final draft.
- Guangjun Wang performed the experiments, prepared figures and/or tables, and approved the final draft.
- Wangbao Gong performed the experiments, prepared figures and/or tables, and approved the final draft.
- Jingjing Tian performed the experiments, authored or reviewed drafts of the article, and approved the final draft.
- Hongyan Li analyzed the data, authored or reviewed drafts of the article, and approved the final draft.
- Kai Zhang analyzed the data, authored or reviewed drafts of the article, and approved the final draft.
- Yun Xia analyzed the data, authored or reviewed drafts of the article, and approved the final draft.
- Zhifei Li conceived and designed the experiments, authored or reviewed drafts of the article, and approved the final draft.

- Jun Xie conceived and designed the experiments, authored or reviewed drafts of the article, and approved the final draft.
- Ermeng Yu conceived and designed the experiments, performed the experiments, authored or reviewed drafts of the article, and approved the final draft.

## Ethics

The following information was supplied relating to ethical approvals (*i.e.*, approving body and any reference numbers):

This study was approved by the Pearl River Fisheries Research Institute, CAFS (LAEC-PRFRI-2021-03-03).

## Data Availability

The raw measurements are available in the Supplemental Files.

## Supplemental Information

Supplemental information for this article can be found online at http://dx.doi.org/10.7717/peerj.15733#supplemental-information.

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
