# Peer review of "Combined effects of dietary faba bean water extract and vitamin K3 on growth performance, textural quality, intestinal characteristics, oxidative and immune responses in grass carp"

_PeerJ, doi:10.7717/peerj.15733_

## Round 0.1 · original submission · Minor Revisions

Expressing the findings more clearly will contribute to the development of scientific content.

Reviewer 1 ·

Basic reporting

Basic reporting
In this manuscript, the authors used two oxidants, faba bean water extract (FBW) and vitamin K3 (VK3) in grass carp. The muscle-hardening effects of FBW and VK3 have been already reported in this and other species, and the main aim of this study is the dose and combined supplementation (the FBW + VK3 group). They showed that the combined supplementation is suitable for commercial application.

Overall, this is a well-conducted study and adds new insights into aquaculture of grass carp. The multiple sets of data are impressive and seem to have captured the characteristics of these experimental groups accurately.

The manuscript is well written, but minor English proofreading is necessary. For example L107 "increased", L223 There "were" no...

Experimental design

1. My biggest concern is the clarification and justification of the dose of FBW and VK3. The authors cited Ma et al. (2020) and Yu et al. (2020) around L100. What was the percentage of FBW and VK3 in these studies? The amount is witten in the first paragraph of Discussion, but this information should be clearly stated in Introduction in comparison of the dose used in this study.

2. Related to the above comment, I wonder why the authors did not include high dose groups of each oxidant, such as 30% FBW and 5% VK3. Otherwise it is difficult to use the word "synergistic effect" since the observed effect may simply be proportional to the oxidant contents in diets. I consider that changes in the "synergistic" group should be compared with 30% FBW and 5% VK3 groups.

Validity of the findings

The experiments and statistical analysis appeared to be conducted appropriately. I found no particular flaw in their data. The use of the term "synergistic" should be reconsidered.

Reviewer 2 ·

Basic reporting

The article is written in a generally understandable language, the target to be achieved and the study design are clearly stated. Findings were given with the help of tables and graphics and statistical interpretations were made. However, in the Methods section, the histopathological examination of the intestines and muscles and the morphometric measurement data were not included in the findings section. It does not seem possible to understand the pictures with the given brief information. It would be more appropriate to add these data and arrange some propositions at the end of the study in line with the questions asked.

Experimental design

It is seen that the methodology of the research was carried out in accordance with the literature and ethical rules. However, it was noted that some of the studies included in the methodology were not included in the results.
Line 195-197: The enzymes analyzed are enzymes from the intestinal contents. Why did you include intestinal tissue in the analysis? Doesn't this affect the results negatively?
Line 203-213: The results of the analyzes described here are not included in the findings and tables.

Validity of the findings

Results
Findings were generally presented in an understandable way and supported by tables and graphs. However, as I mentioned in the method section, histopathological and morphometric examination results of the muscles and intestines were not included. These results need to be added or removed from the methodology and discussion.
Line 292-296: Where are the muscle fiber measurements and intestinal histopathological examination results? There are only pictures, no measurement charts
You need to put the measurement results, if any, or the scoring results, if any. It is not possible to understand what you are saying with just a picture.

Additional comments

Discussion

Considering the differences in the findings, the discussion was made in accordance with the existing literature and suggestions were made. However, revealing the reasons for some propositions will increase the emphasis.

Line 322-324: It had no effect in previous doses, and it has no effect in the current doses you use as high doses. How did you conclude that it would be effective at higher doses?

A few minor typos have been flagged in the article: Line 317, 340, 439

Can be published after the above-mentioned corrections are made (morphometric measurement and histopathological findings table)

Annotated reviews are not available for download in order to protect the identity of reviewers who chose to remain anonymous.

---

## Round 0.2 · accepted · Accept

In line with the reviews of valuable reviewers, I think that the article provides academic qualifications.

Reviewer 1 ·

Basic reporting

The authors appropriately addressed my concerns.

Experimental design

No more concerns

Validity of the findings

No more concerns

Additional comments

No more concerns

Reviewer 2 ·

Basic reporting

The article is written in a generally understandable language, the goal to be achieved and the study design are clearly stated. Findings were given with the help of tables and graphics and statistical interpretations were made. In the methodology, findings and discussion sections, positive feedback was given to the criticisms and the desired corrections were made. It is appropriate to publish the article

Experimental design

It is seen that the methodology of the research is carried out in accordance with the literature and ethical rules.

Validity of the findings

The findings were presented in a generally understandable way and supported by tables and graphs. The new regulations were also found appropriate.

Additional comments

In the article, the corrections requested to be made in the previous revision have been made appropriately. The publication of the article was deemed appropriate.